# Isolation of *Clostridioides difficile* from a Large Animal Veterinary Teaching Hospital Environment

**DOI:** 10.3390/ani15182703

**Published:** 2025-09-15

**Authors:** Alexandre S. Borges, Luiza S. Zakia, Serena Yu, Michael G. Surette, Luis G. Arroyo

**Affiliations:** 1Departments of Clinical Studies, Ontario Veterinary College, University of Guelph, Guelph, ON N1G 2W1, Canada; alexandre.s.borges@unesp.br (A.S.B.); lstachew@uoguelph.ca (L.S.Z.); syu11@uoguelph.ca (S.Y.); 2Department of Medicine, Farncombe Family Digestive Health Research Institute, McMaster University, Hamilton, ON L8S 4L8, Canada; surette@mcmaster.ca

**Keywords:** equine, spore, anaerobic, nosocomial

## Abstract

*Clostridioides difficile* is considered an important cause of enterocolitis in horses. The risk of *C. difficile* nosocomial acquired infections remains largely unknown, and only a few studies surveyed the environmental prevalence of *C. difficile* in these facilities, however. This study found that the prevalence of toxigenic *C. difficile* in a large animal veterinary hospital is low; however, the risk of nosocomially acquired infection in hospitalized horses remains to be determined. Periodic surveillance is important; nevertheless, to monitor the environmental contamination, the detection of emerging strain lineages, and their antimicrobial resistance profiles.

## 1. Introduction

*Clostridioides difficile* (formerly *Clostridium difficile*) is a Gram-positive, spore-forming anaerobic bacterium, and a well-known etiological agent of gastroenteric disease in several species [1,2]. *C. difficile*-associated infection has been reported in domestic and non-domestic animals, including horses. Despite many decades since *C. difficile* was associated with gastrointestinal disease in horses, epidemiological data remain limited [3,4,5]. The clinical disease described in horses includes diarrhea of various degrees of severity, dehydration, toxemia, and colic [4,6]. The source of infection in animals is mostly unknown, but it is presumed that animals ingest *C. difficile* spores widely distributed in their environment [4,7]. The ability of *C. difficile* to sporulate allows these organisms to survive in a wide range of environments, under adverse conditions, and for prolonged periods [8].

Human hospitals and other healthcare facilities’ environments are commonly contaminated with *C. difficile* spores, which are also frequently resistant to many routinely used cleaning disinfectants [9]. Consequently, hospital-acquired *C. difficile* infection in humans is amongst the most important nosocomial infections and a major cause of morbidity and mortality. In the human healthcare system, the incidence of hospital-acquired *C. difficile* infections remains high, despite the suggestion that appropriate prevention strategies may reduce the risk of infection [10]. Conversely, hospital-acquired *C. difficile* infections and outbreaks in veterinary hospitals are rare and only sporadically reported [1,11,12]. Further factors strongly associated with the risk of developing nosocomial infections in humans, such as antimicrobial use, have not been confirmed in veterinary hospitals. Although *C. difficile* is prevalent in the veterinary hospital’s environment, the role of the environment as a source of infection for animals or the potential for zoonotic risk is largely unknown [11,12,13,14,15]. Nevertheless, environmental infectious disease surveillance in human and veterinary hospitals is paramount in order to determine the level of environmental contamination and effectiveness of cleaning and disinfection protocols [10]. Therefore, the aim of this study was to determine the prevalence of *C. difficile* in the Ontario Veterinary College large animal clinic (OVC-LAC) environment and to characterize the recovered isolates. It was hypothesized that *C. difficile* is present in the hospital environment of the large animal hospital of the OVC-LAC.

## 2. Materials and Methods

### 2.1. Sample Collection

Environmental samples were collected from several surface areas of the Ontario Veterinary College-Large Animal Hospital, University of Guelph. The sampling was performed over the course of two days (September 2019), and the number of samples collected per location was arbitrarily established according to the frequency of daily animal and human traffic [10]. Sampled surfaces included stall walls and floors, hallways, isolation units, sinks, doors, and computer keyboards, and the sampling took place at the end of the working day before cleaning the hospital. The hospital’s environmental cleaning protocol includes daily scrubbing of the stalls’ walls and floors with an accelerated hydrogen peroxide solution, followed by thorough hosing. A surface area of 40 × 40 cm per site was wiped using sterile cellulose sponges on a quick-release handle pre-moistened in Dey-Engley neutralizing buffer for environmental sampling (Solar-Cult^®^ Sponge-Stick, Solar Biologics, Sigma Aldrich. Oakville, ON, Canada). Sponges were placed in sterile zip-lock bags, and the handle was detached and discarded. Then, 40 mL of sterile water was added, and each sponge was vigorously massaged manually for 2 min. The sponges were squeezed inside the bags in order to extract the liquid and debris, and the yielded fluid (approx. 50 mL) was transferred into a sterile plastic conical tube. Conical tubes were centrifuged at 3000× *g* for 25 min, and the supernatant was discarded. Pellets were re-suspended in approx. 600 µL of sterile water was aliquoted (2 × 250 µL aliquots) for anaerobic culture.

### 2.2. C. difficile Cultivation

An aliquot of 250 µL was transferred into 9 mL non-selective Fructose Broth supplemented with 0.1% sodium Taurocholate (40 g/L proteose peptone, 5 g/L disodium hydrogen phosphate, 1 g/L potassium dihydrogen phosphate, 0.1 g/L magnesium sulfate, 2 g/L sodium chloride, 6 g/L fructose, 1 g/L sodium taurocholate, pH 7.4) (Thermo Fisher Scientific, Mississauga, ON, Canada); and anaerobically incubated at 37 °C for 8 days. Then 2 mL of cultured broth was transferred into a sterile glass tube and mixed with an equal volume of absolute ethanol and incubated at room temperature for 1 h. Tubes were centrifuged at 3000× *g* for 15 min, the supernatants were discarded, and the pellets were plated onto solid selective *C. difficile* CDMN media (*C. difficile* moxalactam norfloxacin Oxoid, Canada). Plates were incubated at 37 °C in an anaerobic chamber for 5 days. Suspected *C. difficile* colonies based on their morphology and typical odor were sub-cultured into Columbia agar plates (Oxoid Canada) and incubated anaerobically for 48 h. *C. difficile* suspected colonies were then screened by using enzymatic hydrolysis of L-proline-naphthylamide (ProDisc Hardy Diagnostics Canada), and the bacterial identification was confirmed by MALDI-TOF (MALDI Biotyper System, Bruker Daltonics, Billerica, MA, USA).

### 2.3. Molecular Analysis

#### 2.3.1. DNA Extraction

DNA was extracted from pure culture colonies (Instagene Matrix—Biorad) and stored at –20 °C until further analysis. Multiplex PCR method for the detection of *C. difficile* toxin A (*tcdA*) and toxin B (*tcdB*) and the binary toxin (*cdtA*⁄*cdtB*) genes was performed as previously described [16]. The DNA was also used for a High Resolution Capillary Gel-Based Electrophoresis PCR-Ribotyping, and *C. difficile* capillary-sequencer-based PCR-ribotyping was automatically analyzed and compared using an online database (https://webribo.ages.at/ accessed on 1 September 2025) [17].

#### 2.3.2. Whole Genome Sequencing Analysis

Whole-genome sequencing libraries were prepared using a miniaturized protocol with the NEBNext Ultra II FS DNA Library Prep Kit for Illumina (NEB, Ipswich, MA, USA) [18]. Barcoded libraries were size-selected using the ProNex^®^ Size-Selective Purification System (Promega, Madison, WI, USA) to enrich for insert sizes of 800–1000 bp and sequenced on an Illumina HiSeq2500 platform in rapid run mode, paired-end 2 × 250 nt, at the Farncombe Metagenomics Facility (Hamilton, ON, Canada). Genome assemblies were carried out with Unicycler [19]. These whole-genome shotgun assemblies have been deposited at DDBJ/ENA/GenBank under BioProject PRJNA1301804.

#### 2.3.3. Multilocus Sequencing Typing Analysis

The Multilocus Sequencing Typing (MLST) of sequenced genomes was performed at PubMLST [20,21]. A core gene SNP phylogenetic tree was constructed from the environmental isolates with 114 *C. difficile* genomes from the RefSeq database (https://www.ncbi.nlm.nih.gov/refseq/ (accessed on 4 August 2024)) (Appendix A). One environmental isolate (ST 2) was excluded because of low-quality genome assembly. All genomes were annotated with bakta v1.5.0 [22] to have consistent gene prediction across all genomes. A core-gene alignment was generated with panaroo v1.5.0 [23] with the following parameters [core --aligner clustal --core_threshold 0.98 --remove-invalid-genes --search_radius 0]. iqtree_v2.2.6 [24] was first run to determine the optimum model [25] and then used to construct the final bootstrapped tree using the following parameters [-m GTR+F+I+R7 -T 20 -bb 1000]. The final tree was visualized and annotated using FigTree v 1.4.4 (https://github.com/rambaut/figtree (accessed on 1 September 2025)).

#### 2.3.4. Antimicrobial Resistance Gene Analysis

Antimicrobial resistance was predicted from sequenced genomes using three programs. The Resistance Gene Identifier (v6.0.050 online tool) at The Comprehensive Antibiotic Resistance Database (CARD, V4.0.01) was used with Perfect and Strict search parameters only [26]. AMRFinderPlus (v4.0.23) was used on a local server (amrfnderplus database version 2025-07-16.1) using each genome.fna file with --nucleotide and --organism *C. difficile* flags [27]. ResFinder (v4.7.2) predictions were run on the online server (https://genepi.dk/resfinder (accessed on 12 June 2025)) with the default parameters [28,29]. There were some discrepancies between the three tools in the predicted nomenclature for aminoglycoside phosphotransferases. Predictions were aligned, and the consensus nomenclature was used for these genes. Briefly, ResFinder gave the same genes *aac(6*′*)-Ie/aph(2*″*)-Ia* and *aph(2*″*)-Ia* predictions, and the latter assignment was removed. AMRFinderPlus and ResFinder *ant(6)-Ia* genes were predicted as *aad(6)* by RGI-CARD, and the former name was used. AMRFinderPlus and ResFinder predicted genes assigned as *aadE* and *ant(6)-Ia*, respectively. These were distinct from the *ant(6)-Ia* genes above; therefore, the *aadE* nomenclature was used. These genes were not found by RGI-CARD. Virulence factor prediction was carried out using MetaVF_toolkit with the VFDB2.0database [30] on a local server using default parameters.

## 3. Results

### 3.1. C. difficile Cultivation

A total of 157 sites were sampled (Table 1) from 2 separate buildings, of which 13 cultured positive for *C. difficile*. 

All the isolates were recovered from the main hospital area, whereas all the samples taken from the equine isolation building were culture-negative.

### 3.2. Molecular Analysis

#### 3.2.1. Ribotyping and Multilocus Sequencing Typing Analysis

These 13 isolates were classified into 10 distinct ribotypes, 7 were PCR positive for genes coding for toxins A (*tcdA*) and B (*tcdB*) (A+B+), 6 non-toxigenic (A-B-), and all were binary toxin negative (CDT-) (Table 1). This was confirmed through whole-genome sequence analysis.

Bacterial genomic sequence of the isolates showed that all 13 identified were within the MLST Clade 1 and represented five different sequence types (ST): ST54, ST26, ST15, ST14 and ST2 (Appendix A). The distribution of these strains in the phylogenetic tree of core gene SNPs was constructed for 12 isolates with 114 *C. difficile* RefSeq genomes, as shown in Figure 1.

Accession numbers, strain types, and clades for the strains are summarized in Appendix A.

#### 3.2.2. Antimicrobial Resistance Gene Analysis

Antimicrobial resistance genes (ARG) were predicted using three bioinformatics tools, and the results are summarized in Table 2, which includes only ARG predicted by at least 2 of the 3 tools.

Notably, *ermB* was predicted in 10 of 12 isolates (83%), *ermQ* in 1 isolate (8%), and *tetM* in 7 of 12 isolates (54%). Six different ARGs conferring aminoglycoside resistance were predicted. *aac(6*′*)-Ie/aph(2*″*)-Ia* (aminoglycoside acetyltransferase), *ant(6)-Ia* (aminoglycoside nucleotidyltransferase), *aph(3*′*)-IIIa* (aminoglycoside phosphotransferase), and *sat4* (streptothricin acetyltransferases) were detected in 33% of isolates). *aadE* (aminoglycoside nucleotidyltransferase) and *aph(2*″*)-If* (aminoglycoside 2″-phosphotransferase, were found in 83 and 25% of isolates, respectively. There was some discrepancy in gene nomenclature between the three tools for these (see Methods), but each identified ARG in Table 2 represents a unique gene sequence. A 23S rRNA methyltransferase (*cfrC*) associated with resistance to linezolid and chloramphenicol in clostridia [31] was identified in 3 isolates (25%). *vanRG and vanXYG* were predicted in 100% *C. difficile* genomes by CARD-RGI, but no vancomycin resistance was observed in any of the isolates. CARD-RGI also predicted a multidrug and toxic compound extrusion (MATE) family *cdeA* in all genomes.

## 4. Discussion

*C. difficile* was recovered from multiple sites of the large animal hospital environment, as expected. The role of *C. difficile* as a cause of enteric disease in some domestic animals had been established [6,32]. In horses, it is considered one of the most important causes of diarrhea, often requiring hospitalization and intensive care [14,33]. Contrary to humans, hospital-acquired *C. difficile* infections are not well documented in veterinary hospitals but have been suspected [11,13]. *C. difficile* is one of the most important causes of hospital-acquired infections and by far the main cause of healthcare-associated diarrhea in human patients [34]. *C. difficile* spores contaminating hospitals and other healthcare facilities constitute an imminent threat to at-risk human patients, such as those on antibiotic therapy, receiving proton pump inhibitors, chemotherapy, and the elderly. The contamination rates reported for human hospitals vary from 9.7% to 58%, depending on the location of the study, sampling and culture methods used, and disinfection protocols [35]. A recent study showed an overall pooled prevalence of *C. difficile* in hospital environments of 14.9%; however, prevalence as high as 51.1% was reported in countries such as India, and as low as 1.6% in the USA [36].

In veterinary medicine, this organism had been previously cultured from our large animal hospital wards and hospital environments in other countries [12,14,37]. The observed prevalence (8.3%) in this study is overall similar to those reported for veterinary hospitals [12,38], however, higher than the prevalence (4.5%) reported for our large animal hospital over 20 years prior [14]. There are several possible explanations for the observed increased prevalence, including a higher environmental contamination. The sampling technique and culture methods may play a role in the isolation rate between studies. Premoistened sponge swabs with neutralizing solution were used to sample the hospital environmental surfaces. The sponge sampling technique used for our study is superior to contact agar plates for the recovery of *C. difficile* from sites in the clinical environment and is the preferred method for routine surface hygiene monitoring in some healthcare facilities [39]. Additionally, enrichment broth, as used for this study, is generally more sensitive than direct agar culture for the isolation of *C. difficile* spores from environmental samples.

A higher prevalence of *C. difficile* has also been reported in small animal veterinary clinics and their environment; however, there is no apparent risk for acquired *C. difficile* infection in these sites either, or otherwise unrecognized/unreported [15,23]. A recent study found *C. difficile* in nearly all samples collected from the shoe soles of veterinarians, veterinary support staff, and veterinary students at the veterinary school campus in Europe [40]. Despite the apparent ubiquitous presence of this organism in veterinary environments, an increase in risks for *C. difficile* infection and disease development has not been demonstrated in animals, including horses. Conversely, in human healthcare facilities, exposure to *C. difficile* commonly occurs in hospitals where spores persist for prolonged periods in the environment and play an important role in the transmission of the disease [41,42].

All the isolates recovered in this study were cultured from samples collected from the main hospital, and no clustering to particular hospital areas was observed. Many isolates were recovered from hallways, however, which may be correlated with high traffic of animals and hospital personnel, and challenges related to adequate cleaning and disinfection of those sites. Similar to our observations, Weese and co-workers reported isolation of *C. difficile* from areas with high animal traffic and with rough, difficult-to-clean surfaces [14]. They also found a geographic association between areas in the large animal hospital where *C. difficile* diarrheic in horses had been hospitalized; however, that was not the case in this study. Diarrheic horses had been admitted into a self-contained isolation unit for over 15 years, which prevents the changes of environmental contamination in the main hospital. Therefore, the increased prevalence of *C. difficile* in this area of the hospital cannot be attributed to contamination by diarrheic horses.

In contrast, *C. difficile* was not recovered from samples collected from the equine isolation unit, which was unexpected since all horses with diarrhea are hospitalized in this building. Although the environmental prevalence of *C. difficile* can be variable among human hospital wards, the physical layout of veterinary hospital wards is not similarly structured [24,43]. Some possible reasons for culture-negative samples from the isolation unit may indicate a more stringent cleaning and disinfection protocols and easier surfaces to clean.

There was a wide variety of ribotypes [10] among the isolates, which was also not unexpected. Contrary to human healthcare facilities, where clonal and virulent strains can be highly prevalent in the environment and cause multiple outbreaks [44], a large heterogeneity of isolates is found in clinical cases and veterinary hospitals and clinics environments [15,45,46]. Ribotype 014 was recovered from 2 samples, and this strain is considered a human hospital-associated lineage but has also been detected in animal samples, especially pigs [47]. Overall, the two most common PCR ribotypes were 014 and 010, similar to a previous study in samples from puddles and/or soil [48]. Although the vast majority of the caseload in this large animal hospital is horses, a variety of other animals, including pigs, are routinely hospitalized. The source of such isolates is unknown, but interestingly, this ribotype was also isolated from a veterinary hospital environment in Spain [12]. In a recent study in horses, RT 014 was the only ribotype isolated that corresponded to an international reference collection [49]. Ribotype 014-020 was the second most common endemic strain isolated from human stool samples in an 8-year-long study in Texas, USA, and had also been frequently isolated from animals in Brazil [50,51]. Culture of *C. difficile* isolates of a particular ribotype in the same area was not observed, and only RT 14 and 10 were isolated in other areas in this study. RT 078, commonly isolated from animals in previous studies, was not isolated from the environment [52].

Seven isolates (53%) were classified as toxigenic based on PCR detection of genes coding for toxin A and B, and none of the isolates contained the binary toxin. Similarly, Weese and co-workers reported that 14/21 (67%) of the environmental isolates recovered were toxigenic [14]. Similarly, in human hospitals, most environmental *C. difficile* isolates recovered are toxigenic [53,54].

Despite the presence of toxigenic strains of *C. difficile* in veterinary hospitals and private clinics, nosocomial acquired infections are rarely reported [11,13]. In contrast to human medicine, nosocomial and community home care-acquired *C. difficile* infections play a major role in the epidemiology of the disease and remain an infection control challenge largely due to environmental spore contamination [55]. Therefore, environmental decontamination of rooms, equipment, and utensils using a sporicidal product and strict compliance with cleaning and disinfection protocols are critical.

All isolates in our study were classified by MLST in Clade 1 and grouped into five distinct sequence types or STs. MLST sorts strains into clades based on common molecular lineages, and five *C. difficile* evolutionary clades have been described up until now. Each clade is composed of STs, and for clade 1, for example, 106 STs had been reported [20,56,57]. Clades are also geographically related to a continent of origin as follows: clade 1 (Europe), clade 2 (North America), clade 3 (potentially Africa), clade 4 (Asia), and clade 5 (Australia) [41]. This typing method has not been applied to *C. difficile* isolates recovered from veterinary hospitals’ environment; therefore, a comparative analysis was not performed. It is interesting that all environmental isolates from this study belonged to Clade 1, whereas most equine infections have been reported to belong predominantly to Clade 5 [58]. Indeed, contemporary *C. difficile* isolates at this hospital from a mare and an infected foal belonged to Clade 5 (Appendix A). If the prevalent environmental isolates within the hospital belong to Clade 1 rather than Clade 5, this may contribute to the apparent low nosocomial infection rate reported in veterinary hospital-acquired infections [11,13]. Interestingly, however, a recent study in humans reported that both toxigenic and nontoxigenic strains of *C. difficile* clade 1 were the most prevalent (131/146, 89.7%) isolates recovered from surfaces and patient fecal samples [59].

Although MLST and ribotyping have similar discriminatory powers, different ribotypes might be seen as a single sequence type by MLST, and vice versa. For example, ribotype 014 falls into a number of sequence types (ST2, ST14, ST50, and ST132) [57]. Remarkably, many *C. difficile* isolates from animal origin had been assigned to ST 11, although the RT grouping within this ST is highly heterogenic [56]. Since there are only a few studies using MLST for typing *C. difficile* isolates from animal origin, the prevalence of STs, including ST54, ST26, ST15, and ST2, is unknown. Although a good association considering PCR ribotyping and strain types of equine origin was previously observed [49].

*C. difficile* resistance to antimicrobials has been reported in humans and animals, including horses [32].

Several mechanisms of resistance have been previously identified in *C. difficile*, including acquisition of genetic elements and alterations of the antibiotic target sites [60]. The high prevalence of ARG on these tested genomes is of interest, particularly since few AMR-genes (i.e., *erm*(A), *erm*(B), *tet*(M), *ant(6)-Ib*, *catD,* and *cfr*(B)) have been characterized in *C. difficile* isolates [61].

CdeA, a multidrug efflux transporter of the multidrug and toxic compound extrusion (MATE) family, had been previously identified in *C. difficile* strains [62]. The prevalence or role of this resistant gene is unknown, however. The 23S rRNA gene mutation and ribosomal proteins have been reported as a mechanism of antimicrobial resistance for Gram-positive bacteria, including *C. difficile* [63]. The detection of the mutation in these isolates, however, does not imply inherited resistance of these isolates. Predicted *vanRG/vanXYG* was found in a high proportion of the isolates with CARD-RGI but not the other ARG prediction tools. It had been shown that the *vanG_Cd_* gene cluster in the *C. difficile* genome is expressed and functional, although the organism remains susceptible to vancomycin [64]. The antimicrobial resistance profiles of *C. difficile* may be associated with the emergence of clonal strains rather than a widely spread carriage of resistant genes amongst highly heterogeneous *C. difficile* isolates; however, this observation warrants further investigation [65].

A limitation of this study was the lack of antibiotic susceptibility testing since AMR gene detection does not confirm phenotypical resistance. Isolates from historical clinical cases were available to determine if the environmental isolates are indistinguishable from those recovered from horses with *C. difficile*-associated diarrhea.

## 5. Conclusions

In summary, the prevalence of toxigenic *C. difficile* in a large animal veterinary hospital remains low; however, the potential role as a source for nosocomial infection and development of clinical disease remains to be determined. Periodic surveillance remains important to monitor the environmental contamination load, the detection of emerging strain lineages, and their antimicrobial resistance profiles.

## Figures and Tables

**Figure 1 animals-15-02703-f001:**
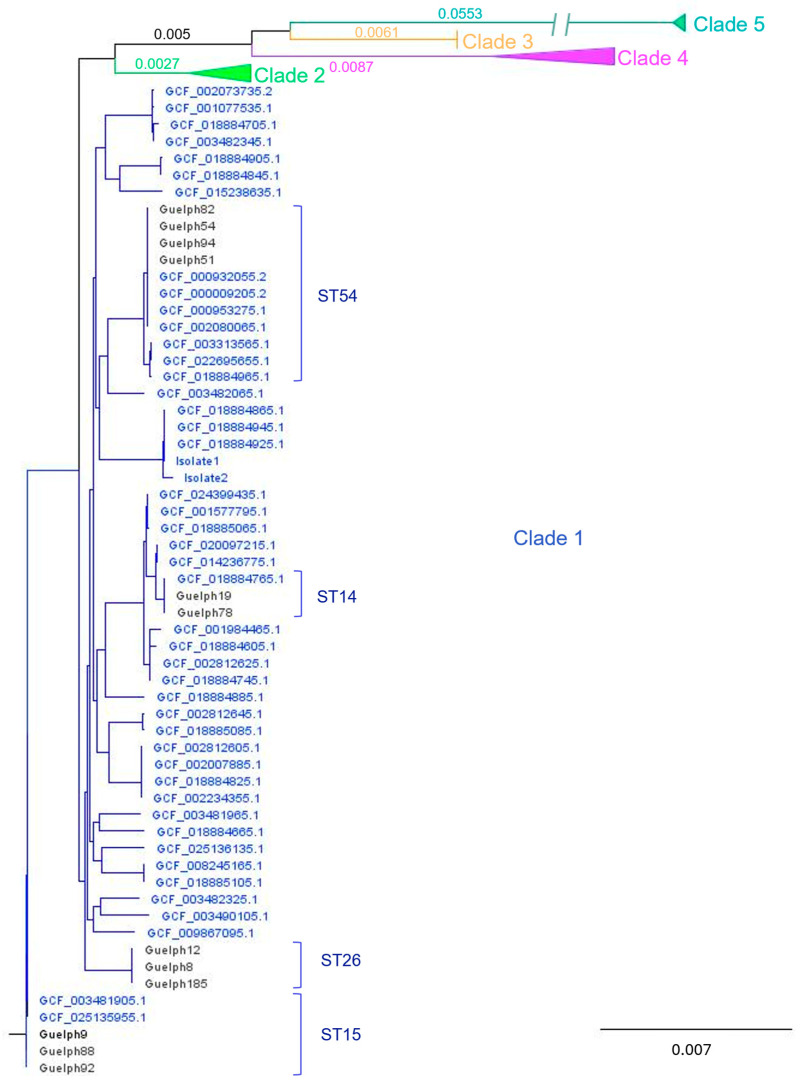
*C. difficile* phylogenetic tree of core gene SNPs for the environmental isolates (designated Guelph followed by a number) with 114 reference genomes (GCF are reference numbers for the NCBI RefSeq Assemblies) spanning clades 1–5. All environmental isolates from this study were in Clade 1, and relevant ST types are indicated. All reference isolates included here were isolated from humans, with the exception of the two ST15 genomes, which were isolated from a sewage treatment facility. To facilitate visualization, Clades 2–5 are collapsed (a full tree is shown in Appendix A). Branch lengths for the clades are indicated. Note that the long branch for Clade 5 is truncated for visualization.

**Table 1 animals-15-02703-t001:** Environmental location and sites sampled from the Large Animal Clinic for *C. difficile* cultivation and PCR analysis.

Location	Site	No. Samples (Positive)	Sample ID	Ribotype	Toxin Profile
Reception Area	floor	2			
Computers	keyboards	4			
Horse stalls	floor	8			
	wall	8 (1)	Guelph54	25360	A^+^B^+^CDT^−^
Horse examination Rooms	floor	6 (1)	Guelph92	010	A^−^B^−^CDT^−^
	wall	6			
Cow stall 1	floor	6			
	wall	4			
Cow stall 2	floor	2 (1)	Guelph19	014/5	A^+^B^+^CDT^−^
Bathroom	floor	1			
Large animal radiology room	floor	2			
Small ruminant stalls	floor	4			
	wall	4			
Barn hallways drains		18 (2)	Guelph88	010	A^−^B^−^CDT^−^
			Guelph1	012	A^+^B^+^CDT^−^
Breezeway	floor	13 (2)	Guelph9	010	A^−^B^−^CDT^−^
			Guelph8	25358	A^−^B^−^CDT^−^
Barn hallways	floor	16 (3)	Guelph185	25365	A^−^B^−^CDT^−^
			Guelph82	25362	A^+^B^+^CDT^−^
			Guelph78	014/5	A^+^B^+^CDT^−^
Surgical suits	floor	4			
Surgical induction room	floor	2 (1)	Guelph94	25364	A^+^B^+^CDT^−^
Ruminant Isolation Unit	floor	2 (1)	Guelph20	449	A^+^B^+^CDT^−^
MRI induction room	floor	1			
	Wall	1			
Nuclear Medicine room	floor	1			
Ruminants ward hallways	floor	2 (1)	Guelph12	25393	A^−^B^−^CDT^−^
Surgical Recovery Room	floor	2			
Equine ward hallways	floor	12			
Equine Isolation-Exam Room	floor	4			
Equine Isolation Hallways	floor	6			
Equine Isolation Stalls	floor	8			
	wall	8			

**Table 2 animals-15-02703-t002:** Multilocus sequence typing, toxin and antimicrobial resistance predictions from whole genome assemblies of *Clostridioides difficile* environmental isolates.

Isolate	ST	MLST Clade	*tcdA*	*tcdB*	*aac(6*′*)-Ie/aph(2*″*)-Ia*	*aadE*	*ant(6)-Ia*	*aph(2*″*)-If*	*aph(3*′*)-IIIa*	*sat4*	*erm(B)*	*erm(Q)*	*tet(M)*	*cfr(C)*	*gyrA_T82A*
Guelph19	14	1	1	1											
Guelph78	14	1	1	1											
Guelph12	15	1				1		1			1		1	1	
Guelph88	15	1				1					1				1
Guelph92	15	1				1					1				1
Guelph8	26	1				1		1			1		1	1	
Guelph9	26	1				1					1				1
Guelph185	26	1				1		1			1		1	1	
Guelph51	54	1	1	1	1	1	1		1	1	2		1		
Guelph54	54	1	1	1	1	1	1		1	1	2		1		
Guelph82	54	1	1	1	1	1	1		1	1	2	1	1		
Guelph94	54	1	1	1	1	1	1		1	1	1		1		
AMRFinder					√	√	√	√	√	√	√	√	√	√	√
Resfinder					√	√	√	√	√		√	√	√	√	
CARD-RGI					√		√	√	√	√	√	√	√		√

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
