# Peer review of "Isolation of *Clostridioides difficile* from a Large Animal Veterinary Teaching Hospital Environment"

_animals, 2025, doi:10.3390/ani15182703_

Round 1

Reviewer 1 Report

Comments and Suggestions for Authors

Dear authors and Editor,

This is a relevant and straightforward study that evaluated the prevalence of C. difficile in a Large Animal Veterinary Hospital, repeating a similar study performed in the same environment 20 years ago. I have only a few minor comments.

Line 19: I am not sure you can conclude that the low prevalence (~8.33%) is unlikely to be an important cause of infection.

More important than the overall 8% prevalence (13/157) is the source of those samples. Were they all from the same room? All from computer keyboards? This information is presented in Table 1, but should be included in the abstract and better emphasized in the discussion.  

The majority of isolates were found in common hospital areas.  Is this less relevant to horses with non-GI diseases because of a healthy protective microbiome?

It is not correct to conclude anything about contamination load, since your methods are not quantitative (selective culture). You can say just “environmental contamination”.

Author Response

Reviewer 1.

This is a relevant and straightforward study that evaluated the prevalence of C. difficile in a Large Animal Veterinary Hospital, repeating a similar study performed in the same environment 20 years ago. I have only a few minor comments.

A: Many thanks for the time reviewing our manuscript. 

Line 19: I am not sure you can conclude that the low prevalence (~8.33%) is unlikely to be an important cause of infection.

A:  Thank you for this comment.  In all fairness I have to data to support this statement.  We don't have cases of nosocomial acquired diarrhea in our hospital and all tested sites of the isolation unit (a separate building) were negative.  I edited the sentence to "soften" the tone.  Additionally, the overall prevalence in this environment is similar to the overall prevalence in the feces of healthy horses. 

More important than the overall 8% prevalence (13/157) is the source of those samples. Were they all from the same room? All from computer keyboards? This information is presented in Table 1, but should be included in the abstract and better emphasized in the discussion.

A: Thank you.  As pointed out, the isolates originated from multiple sites.  Added to the abstract. 

The majority of isolates were found in common hospital areas.  Is this less relevant to horses with non-GI diseases because of a healthy protective microbiome?

A:  That is likely the case.  At least animals in this area does not house the horses with diarrhea or GI disease as pointed out.  

It is not correct to conclude anything about contamination load, since your methods are not quantitative (selective culture). You can say just “environmental contamination”.

A: Thank for this observation.  Edited. 

Reviewer 2 Report

Comments and Suggestions for Authors

The present study was conducted with the objective of investigating the prevalence of C. difficile strains isolated from veterinary animal hospital environments. The characterisation of the obtained isolates involved the detection of C. difficile toxin genes, PCR-ribotyping, MLST and antimicrobial resistance profiles.

Line 22: replace resistant with resistance

It is imperative that the abstract comprises the information of PCR ribotyping and the dominant RT (with particular reference to the virulent RT).

It is imperative that the methods employed to achieve results in the abstract be rewritten in a more imperative manner.

Line 45: correct, Gram-positive bacteria

Line 91a and 101: correct, 3,000 xg

Line 114: the genes must be italicized.

Line 141: correct, Antimicrobial Resistance Gene

In the section designated as "Results", the results obtained must be accompanied by a subtitle. For instance, the prevalence of C. difficile and the antimicrobial resistance profiles of C. difficile.

Line 188: Please could you provide further clarification as to the meaning of 'Ü' in Table 2? It would be much appreciated if you could include an abbreviation of it in the table.

In the reference list, the name of bacteria must be italic, please check and correct.

Author Response

Reviewer 2.

The present study was conducted with the objective of investigating the prevalence of C. difficile strains isolated from veterinary animal hospital environments. The characterisation of the obtained isolates involved the detection of C. difficile toxin genes, PCR-ribotyping, MLST and antimicrobial resistance profiles.

Line 22: replace resistant with resistance

A: Corrected.

It is imperative that the abstract comprises the information of PCR ribotyping and the dominant RT (with particular reference to the virulent RT).

A:  Added.

It is imperative that the methods employed to achieve results in the abstract be rewritten in a more imperative manner.

A: Edited.

Line 45: correct, Gram-positive bacteria

A: Corrected.

Line 91a and 101: correct, 3,000 xg

A: Corrected.

Line 114: the genes must be italicized.

A: Corrected.

Line 141: correct, Antimicrobial Resistance Gene

A:  Corrected.

In the section designated as "Results", the results obtained must be accompanied by a subtitle. For instance, the prevalence of C. difficile and the antimicrobial resistance profiles of C. difficile.

A:  Added.

Line 188: Please could you provide further clarification as to the meaning of 'Ü' in Table 2? It would be much appreciated if you could include an abbreviation of it in the table.

A: This is a computer error, supposed to be a check mark: Ö.  Corrected.

In the reference list, the name of bacteria must be italic, please check and correct.

A: Corrected.

Reviewer 3 Report

Comments and Suggestions for Authors

Nice work. 

Line 2 – I hope the title is not with hyphens. This is how I see it:  Ani-Mal-Vet

Line 19- therefore (spelling)

Line 95 – Is there a vendor for the fructose broth?

Line 98-99 – I hope this anaerobic incubation

Line 104 – How did you suspect some colonies were C. diff colonies? What is the morphology of the colonies?

Table 2 – What are the meanings of 1,2, and ü ?

Line 219-221 – It seems like the prevalence is not changed over the last 20 years. Were the detection methods and the sanitizing SOPs the same during these 20 years?

Line 295- Authors have mentioned about lack of phenotypic AMR results. The manuscript would be more informative if you have AMR results for phenotypic resistance as well.

Author Response

Reviewer 3.

Nice work.

A: Thank you.  We love working with these bacteria.

Line 2 – I hope the title is not with hyphens. This is how I see it:  Ani-Mal-Vet

A: Correct. It’s not hyphenated.  

Line 19- therefore (spelling)

A: Corrected

Line 95 – Is there a vendor for the fructose broth?

A: We prepared the broth in the laboratory using ingredients all purchased from “Thermo Fisher Scientific, ON, Canada”.  This information was added.

Line 98-99 – I hope this anaerobic incubation

A: added “anaerobically”

Line 104 – How did you suspect some colonies were C. diff colonies? What is the morphology of the colonies?

A: Correct.  Morphology and smell. Added to the text.

Table 2 – What are the meanings of 1,2, and ü ?

A: Sorry, supposed to be a check mark: Ö.  Corrected.

Line 219-221 – It seems like the prevalence is not changed over the last 20 years. Were the detection methods and the sanitizing SOPs the same during these 20 years?

A: Great question. Both changed.  Weese et al., used RODAC plates while we used enrichment both and hospital environment disinfectants had been changed, yet similar recovery rate.

Line 295- Authors have mentioned about lack of phenotypic AMR results. The manuscript would be more informative if you have AMR results for phenotypic resistance as well.

A: Fully agreed.  This would be even more interesting if we compare that to isolates from clinical cases.  At this point we chose to not purse further testing.  We hope this is fine with the reviewer.